# Pharmaceutical Enterprises’ R&D Innovation Cooperation Moran Strategy When Considering Tax Incentives

**DOI:** 10.3390/ijerph192215197

**Published:** 2022-11-17

**Authors:** Yanping Xu, Lilong Zhu

**Affiliations:** 1School of Business, Shandong Normal University, Jinan 250014, China; 2Quality Research Center, Shandong Normal University, Jinan 250014, China

**Keywords:** pharmaceutical enterprises, tax incentives, industry cooperation, R&D innovation, Moran process evolutionary game

## Abstract

Drug R&D innovation contributes to the high-quality development of the pharmaceutical industry, which is related to people’s life and health, economic development, and social stability. Tax incentives and industry cooperation are conducive to promoting pharmaceutical enterprises’ innovation. Therefore, this paper constructs a Moran process evolutionary game model and analyzes the evolutionary trajectory of *N* pharmaceutical enterprises’ drug R&D innovation strategic choice and considers the choice of R&D innovation strategy and non-R&D innovation strategy. We obtain the conditions for the two strategies to achieve evolutionary stability under the dominance of external factors, the dominance of expected revenue, and the dominance of super expected revenue. The evolutionary process is simulated by MATLAB 2021b. The results show that, firstly, when the number of pharmaceutical enterprises is higher than a threshold, the market is conducive to pharmaceutical enterprises choosing an R&D innovation strategy. Secondly, the higher the tax incentives, the higher the probability of pharmaceutical enterprises choosing an R&D innovation strategy. Thirdly, when the R&D success rate increases, pharmaceutical enterprises gradually change from choosing a non-R&D innovation strategy to choosing an R&D innovation strategy. Fourthly, the threshold of strategy change of pharmaceutical enterprises is the same under the dominance of expected revenue and super expected revenue. This paper puts forward some countermeasures and suggestions for promoting the R&D innovation of pharmaceutical enterprises in practice.

## 1. Introduction

It is important to promote pharmaceutical enterprises’ drug R&D for people’s life and health, economic growth, and social stability. Drug R&D innovation refers to the new chemical structure, therapeutic use, or improvement of drug efficacy. The innovation of pharmaceutical enterprises can effectively improve the drug quality. In recent years, pharmaceutical enterprises have paid more attention to R&D innovation. In March 2022, “Fierce Biotech” released the ranking of the world’s top ten R&D investment pharmaceutical companies in 2021. In 2021, Roche’s R&D budget was USD 16.1 billion, up 14% from the previous year, making it the world’s biggest spender on R&D. As a pharmaceutical enterprise based on R&D, Pfizer attached importance to the investment in drug R&D and the improvement of the R&D level. In 2021, Pfizer’s R&D budget reached USD 13.8 billion, up 47% from the previous year. With the increase in R&D investment, the efficiency of Pfizer’s innovative drug development has also been significantly improved, with the success rate of phase II clinical product development reaching 52% and phase III clinical product development reaching 85%. The high level of R&D also brings huge profits to Pfizer.

Drug R&D innovation has entered a new era. To promote the development of the pharmaceutical industry, various countries have issued a series of preferential tax policies and subsidy policies to promote drug R&D, and encouraged pharmaceutical enterprises to actively cooperate with universities and scientific research institutes to improve the technical level of drug R&D. However, there is still the problem of “emphasizing generic drugs and despising original drugs” in drug production.

Therefore, based on the preferential tax policies of government departments and the cooperation mechanism of the industry, this paper constructs a stochastic evolutionary game model of the Moran process. By studying the strategic choice conditions of pharmaceutical enterprises, it aims to solve the following three problems. Firstly, how do the number of pharmaceutical enterprises in the market, government tax incentives, and R&D success rates comprehensively affect the innovation strategic choice? Secondly, what is the difference between the dominance of external factors, the dominance of expected revenue, and the dominance of super expected revenue on the strategic choices of pharmaceutical enterprises? Thirdly, under different external environments, how can pharmaceutical enterprises be encouraged to choose an R&D innovation strategy?

The remainder of this paper is arranged as follows. Section 2 combs and reviews the relevant literature, Section 3 constructs the Moran process evolutionary game model by considering tax incentives and industry cooperation, Section 4 solves the Moran process, Section 5 analyzes the rooting probability of the two strategies in different situations, Section 6 presents the simulation analysis with MATLAB 2021b, and Section 7 discusses the conclusions.

## 2. Literature Review

### 2.1. R&D Innovation and Government Incentives

Drug R&D innovation is the driving force for the development of the pharmaceutical industry and provides better solutions for improving the drugs’ quality. Drug R&D have the characteristics of high cost, a long clinical trial cycle, and strict production standards, which pose great challenges to the operation of small- and medium-sized pharmaceutical enterprises [1]. The outbreak of COVID-19 has led to the emergence of new public health problems [2] and pharmaceutical enterprises need to provide strong guarantees for drugs and other materials [3]. Therefore, the new drugs’ R&D becomes very important. The government can guide and encourage pharmaceutical enterprises to innovate through policy tools, and government subsidies are an important determinant in the innovation process of pharmaceutical enterprises [4]. The government incentives can effectively promote the production activities of enterprises [5]. China, Singapore, India, Ireland, and other countries also offer attractive tax incentives for pharmaceutical manufacturing [6]. Tax incentives can directly or indirectly make up for the lack of innovation investment of enterprises [7] and reduce the R&D costs [8]. Governments can incentivize R&D by providing tax deductions for output-related income from R&D or related activities [9]. The uncertainty of the market may lead to the risk of incentive failure [10]. The government should set up reasonable preferential tax policies to avoid certain risk areas [11] and carry out strict supervision on pharmaceutical enterprises [12].

### 2.2. R&D Innovation Cooperation

The innovation network of industry cooperation is an important part of the national innovation system [13]. Industrial cooperation refers to cooperation between pharmaceutical companies or cooperation with universities and R&D institutions. Industry cooperation is recognized as an effective model for technological innovation, which helps SMEs to seek opportunities from technological development and achieve sustainable competitiveness [14]. In recent years, enterprises have begun to cooperate with other enterprises and scientific research institutions [15] to effectively alleviate the problems existing in enterprise internal innovation [16] and promote innovation in the industry [17]. The degree of enterprise innovation is affected by the cooperation with universities and other scientific research institutions, and knowledge plays a moderating role in the relationship between industry cooperation and product innovation [18]. The number of scientific research institutions has a positive impact on innovation performance [19]. The improvement of product quality requires the joint participation of multiple subjects [20], and the cooperation of the R&D process is crucial to increase enterprise return and maintain technological competitiveness [21].

### 2.3. Moran Strategy Analysis

The Moran process of limited groups is a random process. Based on bounded rationality and limited information, the evolutionary dynamics of individual selection strategies are analyzed to explore the change rules of group behavior [22]. In the Moran process of frequency dependence, the expected return function of each strategy subject under strong and weak selection conditions is obtained, and the probability of population change in the interval before and after is deduced. The evolution stability strategy is analyzed by using the transition probability matrix of the Markov chain and the dominant probability theory. The Moran process considers the number of individuals in the population before the change, and individuals can make a strategic choice based on the information about the returns and the current overall behavior [23]. The combination of randomness and group selection can promote the evolution of cooperation in the aggregate population. Any proportion of cooperation can be sustained [24]. Evolutionary game theory about finite groups is widely used in sociology, economics, management, etc., for example, by building the Moran process game to predict strategies that evolve in groups of players [25], studying the results of stochastic evolutionary game dynamics combining imitation update rules and average payoff-driven update rules [26], and solving the problem of strategy choice for interaction among polluting enterprises [27].

To sum up, the existing literature mainly discusses the impact of tax incentives on innovation or the impact of industry cooperation on innovation. There is still a lack of analysis of pharmaceutical enterprises’ R&D innovation strategic choices through the Moran process, and how tax incentives and industry cooperation can work together to promote innovation.

Therefore, compared with previous studies, this paper is mainly different in the following three aspects. Firstly, a Moran process game model is constructed for pharmaceutical enterprises’ R&D innovation strategy, comprehensively considering the impact of tax incentives and industry cooperation on pharmaceutical enterprises’ strategic choices. Secondly, it analyzes the impact of external factors, expected revenue, and super expected revenue on the strategic choice of pharmaceutical enterprises, and calculates and solves the conditions for the strategy to take root under the dominance of different conditions. Thirdly, this paper use MATLAB 2021b to simulate the strategy changes and the strategy stability conditions under different conditions.

## 3. Model Hypotheses and Construction

This paper analyzes the evolution of R&D innovation strategies and non-R&D innovation strategies by constructing a Moran process. The hypotheses in this paper are as follows.

H1 In the process of drug R&D innovation, the stakeholders of the game are pharmaceutical enterprises. The feasible strategic choices of pharmaceutical enterprises are (R&D innovation strategy, non-R&D innovation strategy), denoted as (I,T). N pharmaceutical enterprises take part in the Moran process. The R&D innovation strategy means that pharmaceutical enterprises increase investment in the R&D of existing drugs to improve the efficacy. The non-R&D innovation strategy refers to the continuous production of drugs according to the existing production methods, without additional R&D investment, technological improvement, etc.

H2 When pharmaceutical enterprises choose the non-R&D innovation strategy, the cost is $C_{pl}$ and the revenue is $R_{T}$. When pharmaceutical enterprises choose the R&D innovation strategy, the cost is $C_{ph}$ and the revenue is $R_{I}$.

H3 When both pharmaceutical enterprises choose the R&D innovation strategy, they can cooperate with universities and R&D institutions, and the R&D success rate is $\eta_{1}$. When one pharmaceutical enterprise chooses the R&D innovation strategy and the other pharmaceutical enterprise chooses the non-R&D innovation strategy, the R&D success rate is $\eta_{2}$.

H4 Government departments promote enterprises’ R&D innovation through tax incentives. For pharmaceutical enterprises that choose the R&D innovation strategy, government departments provide preferential tax exemptions. For pharmaceutical companies that choose the non-R&D innovation strategy, they need to pay this part of the tax. The preferential tax is $\lambda R_{T}$.

The revenues matrix for the strategic choice is shown in Table 1.

## 4. The Frequency of Limited Pharmaceutical Enterprises Depends on the Moran Process

According to the revenues matrix in Table 1, the expected revenues of choosing the R&D innovation strategy and the non-R&D innovation strategy were calculated. Among the N pharmaceutical enterprises, if i pharmaceutical enterprises choose the R&D innovation strategy, the expected revenues of the R&D innovation strategy and the non-R&D innovation strategy are $E_{i}^{I}$ and $E_{i}^{T}$:(1)$$E_{i}^{I}=\frac{i-1}{N-1}E^{II}+\frac{N-i}{N-1}E^{IT}=\frac{i-1}{N-1}\left(\eta_{1}R_{I}-C_{ph}\right)+\frac{N-i}{N-1}\left(\eta_{2}R_{I}-C_{ph}\right),\;i=1,2,\cdots ,N-1$$
(2)$$E_{i}^{T}=\frac{i}{N-1}E^{II}+\frac{N-i-1}{N-1}E^{TT}=\frac{i}{N-1}\left(R_{T}-\lambda R_{I}-C_{pl}\right)+\frac{N-i-1}{N-1}\left(R_{T}-\lambda R_{I}-C_{pl}\right),\;i=1,2,\cdots ,N-1$$

In addition to revenues, external factors have an impact on the strategic choice of pharmaceutical enterprises. By introducing the selection strength, ξ,ξ∈(0,1], and the fitness functions of strategy I and strategy T of pharmaceutical enterprises are respectively constructed.
(3)$$e_{i}^{I}=1-\xi +\xi E_{i}^{I},e_{i}^{T}=1-\xi +\xi E_{i}^{T},\xi \in \left[0,1\right]$$

When further considering the strategic choice of pharmaceutical enterprises under the dominance of super expected revenue, the ordinary linear fitness relationship cannot satisfy the selection process. When ξ>1, the effect function is an exponential nonlinear function.
(4)$$e_{i}^{I}=e^{\xi E_{i}^{I}},e_{i}^{T}=e^{\xi E_{i}^{T}},\xi >1$$

Based on the Moran process, the probability of adding a pharmaceutical enterprise that chooses an R&D innovation strategy is $\frac{ie_{i}^{I}}{ie_{i}^{I}+\left(N-i\right)e_{i}^{T}}N$. At each stage, the number of pharmaceutical enterprises that choose the R&D innovation strategy either increase by one, decrease by one, or remain unchanged. Therefore, the probability transition matrix of the Moran process can be described as a tridiagonal matrix, and the three elements of the diagonal are as follows:(5)$$Z_{i,i+1}=\frac{ie_{i}^{I}}{ie_{i}^{I}+\left(N-i\right)e_{i}^{T}}\times \frac{N-i}{N}$$
(6)$$Z_{i,i-1}=\frac{\left(N-i\right)e_{i}^{T}}{ie_{i}^{I}+\left(N-i\right)e_{i}^{T}}\times \frac{i}{N}$$
(7)$$Z_{i,i}=1-Z_{i,i+1}-Z_{i,i-1}$$

In the process of pharmaceutical enterprise evolution, the probability of the R&D innovation strategy transitioning from state i to another state is 0, and the one-step Markov transition matrix of the Moran process is as follows:(8)$$\left(\begin{array}{ccccccc} \rho_{0,0} & \rho_{0,1} & 0 & \cdots & 0 & 0 & 0\\ \rho_{1,0} & \rho_{1,1} & \rho_{1,2} & \cdots & 0 & 0 & 0\\ 0 & \rho_{2,1} & \rho_{2,2} & \cdots & 0 & 0 & 0\\ \vdots & \vdots & \vdots & \ddots & \vdots & \vdots & \vdots \\ 0 & 0 & 0 & \cdots & \rho_{N-1,N-2} & \rho_{N-1,N-1} & \rho_{N-1,N}\\ 0 & 0 & 0 & \cdots & 0 & \rho_{N,N-1} & \rho_{N,N} \end{array}\right)$$

The Moran process has two stable states: i=N, where all pharmaceutical enterprises choose the R&D innovation strategy, and i=0, where all pharmaceutical enterprises choose the non-R&D innovation strategies. The probability that the R&D innovation strategy and the non-R&D innovation strategy reach stability is calculated.

Let $q_{i}$ denote the probability of evolving from the initial state of i pharmaceutical enterprises choosing the R&D innovation strategy to the final state of N pharmaceutical enterprises choosing the R&D innovation strategy. It can be obtained from the total probability theorem, as:(9)$$\left\{ \begin{array}{l} q_{0}=0\\ q_{i}=Z_{i,i-1}q_{i-1}+Z_{i,i}q_{i}+Z_{i,i+1}q_{i+1},i=1,2,\cdots ,N-1\\ q_{N}=1 \end{array}\right.$$

Assuming $\varphi_{i}=q_{i}-q_{i-1}$, $\chi_{i}=\frac{Z_{i,i-1}}{Z_{i,i+1}}$, we can solve the following formula:(10)$$\left\{ \begin{array}{l} \varphi_{2}=\chi_{1}\varphi_{1}\\ \varphi_{3}=\chi_{2}\varphi_{2}\\ \cdots \\ \varphi_{i}=\chi_{i-1}\varphi_{i-1} \end{array}\right.$$

Since $\sum_{i-1}^{N}\varphi_{i}=1$, it can be found by the recursive formula:(11)$$q_{1}=\frac{1}{1+\sum_{k=1}^{N-1}\prod_{n=1}^{k}\chi_{n}}$$

Since $\varphi_{i}=\chi_{i-1}\varphi_{i-1}$, $q_{i+1}=\chi_{i}\varphi_{i}+q_{i}$, $q_{i}$ can be solved, as:(12)$$q_{i}=q_{1}\left(1+\sum_{k=1}^{i-1}\prod_{n=1}^{k}\chi_{n}\right)=\frac{1+\sum_{k=1}^{i-1}\prod_{n=1}^{k}\frac{e_{n}^{T}}{e_{n}^{I}}}{1+\sum_{k=1}^{N-1}\prod_{n=1}^{k}\frac{e_{n}^{T}}{e_{n}^{I}}}$$

When only one pharmaceutical enterprise chooses the R&D innovation strategy at the beginning, the probability of the R&D innovation strategy finally stabilizing is $\rho_{I}$:(13)$$\rho_{I}=q_{1}=\frac{1}{1+\sum_{k=1}^{N-1}\prod_{n=1}^{k}\frac{e_{n}^{T}}{e_{n}^{I}}}$$

When only one pharmaceutical enterprise chooses the non-R&D innovation strategy at the beginning, the probability of the non-R&D innovation strategy finally stabilizing is $\rho_{T}$:(14)$$\rho_{T}=1-q_{N-1}=\frac{1}{1+\sum_{k=1}^{N-1}\prod_{i=k}^{N-1}\frac{e_{i}^{I}}{e_{i}^{T}}}$$

The higher the probability of the fixed point, the more likely the strategy is to be evolutionarily stable. When $\rho_{I}>\rho_{T}$, the R&D innovation strategy is more likely to be evolutionarily stable.

## 5. Results’ Analysis

### 5.1. Decision Analysis under the Dominance of External Factors

External factors such as public emergencies and policy changes cause the effect function of pharmaceutical enterprises to approach a certain fixed value. When external factors dominate the decision-making of pharmaceutical enterprises, it is a weak selection process, ξ→0. We solved the Taylor expansion of Equations (13) and (14) at ξ→0.
(15)$$\rho_{I}=\frac{1}{1+\sum_{k=1}^{N-1}\exp(\xi \sum_{n=1}^{k}E_{n}^{T}-E_{n}^{I})}\approx \frac{1}{N}+\frac{\xi }{6N}\left(\alpha +N\beta \right)$$
(16)$$\rho_{T}=\frac{1}{1+\sum_{k=1}^{N-1}\exp(\xi \sum_{n=1}^{k}E_{n}^{I}-E_{n}^{T})}\approx \frac{1}{N}+\frac{\xi }{6N}\left(\gamma +N\delta \right)$$

By calculating this, we can obtain: $\alpha =\left(-2\eta_{1}-\eta_{2}\right)R_{I}+3\left(1-\lambda \right)R_{T}+3C_{ph}-3C_{pl}$, $\beta =\left(\eta_{1}+2\eta_{2}\right)R_{I}-3\left(1-\lambda \right)R_{T}-3C_{ph}+3C_{pl}$, $\gamma =\left(4\eta_{1}-\eta_{2}\right)R_{I}-3\left(1-\lambda \right)R_{T}-3C_{ph}+3C_{pl}$, $\delta =\left(-2\eta_{1}-\eta_{2}\right)R_{I}+3\left(1-\lambda \right)R_{T}+3C_{ph}-3C_{pl}$.

Using the research of Taylor et al. [28], this paper studies the strategic choice of pharmaceutical enterprises based on the fixed-point probability $\frac{1}{N}$. When $\rho_{I}>\frac{1}{N}$, the group supports the R&D innovation strategy to replace the non-R&D innovation strategy; when $\rho_{I}<\frac{1}{N}$, the group supports the non-R&D innovation strategy to replace the R&D innovation strategy.

**Proposition 1:** 
*Under the dominance of external factors, when $\eta_{1}>\eta_{2}$, the following condition is favorable for the R&D innovation strategy to replace the non-R&D innovation strategy:*



(17)
$$\eta_{2}R_{I}-C_{ph}-(R_{T}-\lambda R_{T}-C_{pl})>0$$


**Proof.** $\rho_{I}>\frac{1}{N}$ is equivalent to y=α+Nβ>0. When Formula (17) is established, $y(2)=\eta_{2}R_{I}-R_{T}+\lambda R_{T}-C_{ph}+C_{pl}>0$.$\frac{\partial y}{\partial N}=\left(\eta_{1}+2\eta_{2}\right)R_{I}-3\left(1-\lambda \right)R_{T}-3C_{ph}+3C_{pl}$, because of $\eta_{1}>\eta_{2}$, so $\frac{\partial y}{\partial N}>0$. The function is an increasing function, so $\rho_{I}>\frac{1}{N}$ can be obtained. □

Proposition 1 shows that in the condition of weak selection, the choice of R&D strategy of pharmaceutical enterprises is jointly affected by many factors, such as the number of enterprises in the group, the success rate of R&D innovation, and government tax subsidies. When the expected revenue of the R&D innovation strategy is significantly higher than that of non-R&D innovation, pharmaceutical enterprises choose the R&D innovation strategy. The improvement of the drug R&D success rate and government tax subsidy rate promotes pharmaceutical enterprises to choose the R&D innovation strategy.

### 5.2. Decision Analysis under the Dominance of Expected Revenue 

When pharmaceutical enterprises make a strategic choice according to expected revenue, it is a strong selection process, ξ=1. External factors have little influence on the strategic choice of pharmaceutical enterprises. By comparing the utility functions of the two strategies, we can judge the choice preference of pharmaceutical enterprises.
(18)$$h_{i}=e_{i}^{I}-e_{i}^{T},\;i=1,2,\cdots ,N-1$$

Substitute Equations (1)–(3) into Equation (16) to obtain $h_{1}$ and $h_{N-1}$:(19)$$h_{1}=e_{1}^{I}-e_{1}^{T}=\eta_{2}R_{I}+\lambda R_{T}+C_{pl}-C_{ph}-R_{T}$$
(20)$$h_{N-1}=e_{N-1}^{I}-e_{N-1}^{T}=\frac{N-2}{N-1}\eta_{1}R_{I}+\frac{1}{N-1}\eta_{2}R_{I}+\lambda R_{T}+C_{pl}-C_{ph}-R_{T}$$

In this paper, it can be concluded that:

When $h_{1}>0$, it means that the number of pharmaceutical enterprises that choose the R&D innovation strategy is increasing, and gradually invades the non-R&D innovation strategy; when $h_{N-1}<0$, it means that the number of pharmaceutical enterprises that choose the R&D innovation strategy is declining, and the non-R&D innovation strategy invades the R&D innovation strategy.If both $h_{1}>0$ and $h_{N-1}>0$ are satisfied, the R&D innovation strategy replaces the non-R&D innovation strategy and gradually becomes an evolutionary stable solution.If both $h_{1}<0$ and $h_{N-1}<0$ are satisfied, the non-R&D innovation strategy replaces the R&D innovation strategy and gradually becomes an evolutionary stable solution.If $h_{1}>0$ and $h_{N-1}<0$, the strategies cannot invade each other, and the two strategies exist at the same time.

**Proposition 2.** 
*When making strategic choices based on expected revenue, the number of pharmaceutical enterprises in the group has a threshold $N_{0}=\frac{2\eta_{1}R_{I}-\eta_{2}R_{I}+\lambda R_{T}-R_{T}-C_{ph}+C_{pl}}{\eta_{1}R_{I}+\lambda R_{T}-R_{T}-C_{ph}+C_{pl}}$. When $N<N_{0}$, the two strategies exist at the same time, and the hybrid strategy becomes an evolutionary stable solution. When $N>N_{0}$, if $\eta_{2}R_{I}+\lambda R_{T}-R_{T}-\left(C_{ph}-C_{pl}\right)>0$, the R&D innovation strategy becomes an evolutionary stable solution, and if $\eta_{2}R_{I}+\lambda R_{T}-R_{T}-\left(C_{ph}-C_{pl}\right)<0$, the non-R&D innovation strategy becomes an evolutionary stable solution.*


**Proof.** According to Formula (21), the symbol of $h_{N-1}$ is the same as $h(N)=\left(E_{N-1}^{I}-E_{N-1}^{T}\right)\left(N-1\right)=(N-2)\eta_{1}R_{I}+\eta_{2}R_{I}+\left(N-1\right)\left(\lambda R_{T}+C_{pl}-C_{ph}-R_{T}\right)$. If $h(N_{0})=0$, we can obtain $N_{0}=\frac{2\eta_{1}R_{I}-\eta_{2}R_{I}+\lambda R_{T}-R_{T}-C_{ph}+C_{pl}}{\eta_{1}R_{I}+\lambda R_{T}-R_{T}-C_{ph}+C_{pl}}$. We take the derivative with respect to h(N), $\frac{\partial h(N)}{\partial N}=\eta_{1}R_{I}+\lambda R_{T}-R_{T}-C_{ph}+C_{pl}$. If $\eta_{2}R_{I}+\lambda R_{T}-R_{T}-C_{ph}+C_{pl}>0$, $h_{1}>0$ and $\frac{\partial h(N)}{\partial N}>0$ can be obtained, it can be inferred that h(N) is an increasing function of N. When $N\leq N_{0}$, *h*(*N*) < 0 and $h_{N-1}<0$ can be obtained; when $N>N_{0}$, h(N)>0 and $h_{N-1}>0$ can be obtained. If $\eta_{2}R_{I}+\lambda R_{T}-R_{T}-C_{ph}+C_{pl}<0$, $h_{1}<0$ and $\frac{\partial h(N)}{\partial N}<0$ can be obtained, and it can be inferred that h(N) is a subtractive function of N. When $N\leq N_{0}$, h(N)>0 and $h_{N-1}>0$ can be obtained; when $N>N_{0}$, h(N)<0 and $h_{N-1}<0$ can be obtained. □

Proposition 2 shows that when the number of pharmaceutical enterprises in the market is low, the pharmaceutical enterprises have the same probability of choosing the two strategies. When the number of pharmaceutical enterprises in the market is high, revenue becomes the dominant factor. When the revenue of the R&D innovation strategy is higher than that of the non-R&D innovation strategy, pharmaceutical enterprises tend to choose the R&D innovation strategy.

### 5.3. Decision Analysis under the Dominance of Super Expected Revenue

If the pharmaceutical enterprises in the group overemphasize the cost-return of production, the decision-making behavior will rapidly magnify the returns of the selection strategy, which is called the super expected revenue, ξ>1. Given the number of pharmaceutical enterprises adopting R&D innovation strategies at a certain time, with the increase of the selection intensity, ξ, the super expected revenue increases at a faster rate. By judging the ratio of the effect functions of the two strategies, the change process of the strategic choice of pharmaceutical enterprises is analyzed:(21)$$f_{i}=\frac{e_{i}^{I}}{e_{i}^{T}},i=1,2,\cdots N-1.$$

Substituting Equations (1)–(3) into Equation (21), we can obtain $f_{i}^{I}$ and $f_{i}^{T}$:(22)$$f_{i}^{I}=\frac{e_{1}^{I}}{e_{1}^{T}}=\frac{\exp\left[\xi \left(\eta_{2}R_{I}-C_{ph}\right)\right]}{\exp\left[\xi \left(R_{T}-\lambda R_{T}-C_{pl}\right)\right]}=\exp\left[\xi \left(\eta_{2}R_{I}-R_{T}+\lambda R_{T}-\left(C_{ph}-C_{pl}\right)\right)\right]$$
(23)$$\begin{array}{ll} f_{i}^{T}=\frac{e_{N-1}^{I}}{e_{N-1}^{T}} & =\frac{\exp\left\{\xi \left[\frac{N-2}{N-1}\left(\eta_{1}R_{I}-C_{ph}\right)+\frac{1}{N-1}\left(\eta_{2}R_{I}-C_{ph}\right)\right]\right\}}{\exp\left[\xi \left(R_{T}-\lambda R_{T}-C_{pl}\right)\right]}\\ & =\exp\left[\xi \left(\frac{N-2}{N-1}\eta_{1}R_{I}+\frac{1}{N-1}\eta_{2}R_{I}-R_{T}+\lambda R_{T}-\left(C_{ph}-C_{pl}\right)\right)\right] \end{array}$$

If $f_{1}^{I}>1$ and $f_{1}^{T}>1$ are established, which are equivalent to $h_{1}>0$ and $h_{N-1}>0$, pharmaceutical enterprises are more inclined to choose the R&D innovation strategy. With the progress of the timestep, the R&D innovation strategy becomes an evolutionary stable solution. When $f_{1}^{I}<1$ and $f_{1}^{T}<1$ are established, which are equivalent to $h_{1}<0$ and $h_{N-1}<0$, the non-R&D innovation strategy gradually replaces the R&D innovation strategy until it finally becomes an evolutionary stable solution.

**Proposition 3.** *When $\eta_{2}R_{I}+\lambda R_{T}-R_{T}-(C_{ph}-C_{pl})>0$, there is a threshold, $N_{1}$, for the number of pharmaceutical enterprises, $N_{1}=N_{0}=\frac{2\eta_{1}R_{I}-\eta_{2}R_{I}+\lambda R_{T}-R_{T}-\left(C_{ph}-C_{pl}\right)}{\eta_{1}R_{I}+\lambda R_{T}-R_{T}-\left(C_{ph}-C_{pl}\right)}$. When $N>N_{1}$, the R&D innovation strategy becomes an evolutionary stable solution; when $N<N_{1}$, the two strategies exist at the same time, and the mixed strategy becomes an evolutionary stable solution. However, the earnings of pharmaceutical enterprises show a trend of exponential increase and decrease with the change of the selection intensity parameter, ξ*.

**Proof.** According to Formulas (19), (20), (22) and (24), it can be known that $f_{1}^{I}=\exp\left[\xi \left(h_{1}\right)\right]$ and $f_{1}^{T}=\exp\left[\xi \left(h_{N-1}\right)\right]$. When $\eta_{2}R_{I}+\lambda R_{T}-R_{T}-(C_{ph}-C_{pl})>0$, $\xi \left(h_{1}\right)>0$ and $f_{1}^{I}>1$ can be obtained. Let $h_{N1-1}=0$, $N_{1}=\frac{2\eta_{1}R_{I}-\eta_{2}R_{I}+\lambda R_{T}-R_{T}-\left(C_{ph}-C_{pl}\right)}{\eta_{1}R_{I}+\lambda R_{T}-R_{T}-\left(C_{ph}-C_{pl}\right)}$. When $N\leq N_{1}$, $h_{N-1}<0$ and $f_{1}^{T}<1$ can be obtained, and the two strategies coexist. When $N>N_{1}$, $h_{N-1}>0$ and $f_{1}^{T}>1$ can be obtained, and pharmaceutical enterprises tend to choose the R&D innovation strategy. Proposition 3 shows that under the condition of super excepted revenue, pharmaceutical enterprises have a threshold for strategic choice, which is the same as the threshold for expected revenue. In this situation, the main factors affecting strategic choice are the number of enterprises and strategy revenues. □

## 6. Simulation Analysis

MATLAB 2021b was used to simulate the impact of various factors on strategic choice. There are pharmaceutical enterprises in the market that produce a certain drug. When the pharmaceutical enterprises choose the non-R&D innovation strategy, the production cost is 13 and the revenue is 22. When the pharmaceutical enterprises choose the drug R&D innovation strategy, the cost is 18 and the revenue is 35. The success rate of drug R&D, government tax incentives, and the number of pharmaceutical companies are variables. By taking different values for the three variables, the evolution trend of strategic choice under the dominance of the three different factors is analyzed.

### 6.1. Impact of the Number of Pharmaceutical Enterprises

Under the dominance of external factors, we analyzed the impact of the number of pharmaceutical enterprises on the strategic choice, and set ξ=0.1, $\eta_{1}=0.8$, and $\eta_{2}=0.4$. Figure 1 shows the changing trend of fixed-point probability, $N\times \rho_{I}$ and $N\times \rho_{T}$, under λ=0.1 and λ=0.3.

Figure 1A shows the impact of the number of pharmaceutical enterprises on the probability of taking root when the government tax subsidy rate is low. When the number of pharmaceutical enterprises in the group is low, the two strategies invade each other, and the mixed strategy becomes an evolutionary stable solution. When the number of pharmaceutical enterprises is high, the non-R&D innovation strategy invades the R&D innovation strategy. As shown in Figure 1B, when the government tax subsidy rate is high and the number of pharmaceutical enterprises is less than the threshold $N^{*}=4$, the two strategies coexist in the group; when 4≤N≤6, the group is more conducive to pharmaceutical enterprises choosing the non-R&D innovation strategy; when N>6, the two strategies cannot invade each other again, and the two strategies coexist.

### 6.2. Impact of R&D Success Rate

Figure 2 shows the impact of the R&D success rates, $\eta_{1}$ and $\eta_{2}$, on the strategic choice of pharmaceutical enterprises, and sets ξ=0.01, λ=0.2, and N=50. The image shows the fixed-point probabilities $N\times \rho_{I}$ and $N\times \rho_{T}$.

It can be seen from Figure 2 that $\rho_{I}$ increases with the increase of $\eta_{1}$ and $\eta_{2}$, and $\rho_{T}$ decreases with the increase of $\eta_{1}$ and $\eta_{2}$. The greater the probability of the R&D success rate, the greater the expected revenue of pharmaceutical enterprises. Pharmaceutical enterprises are more inclined to choose the R&D innovation strategy. The R&D innovation strategy is easier to take root in the group and gradually becomes an evolutionary stable solution.

### 6.3. Impact of the Number of Pharmaceutical Enterprises and Tax Incentives

Under the dominance of expected revenue, ξ=1, we analyzed the impact of the number of pharmaceutical enterprises and tax incentives on strategic choice. Let $\eta_{2}=0.3$, and Figure 3 shows the changing trend of h under $\eta_{1}=0.7$ and $\eta_{1}=0.5$.

It can be seen from Figure 3A that when the tax incentives are high, there is a threshold for the number of pharmaceutical enterprises. When the number of pharmaceutical enterprises in the group is lower than the threshold, the non-R&D innovation strategy becomes an evolutionary stable solution. When the number of pharmaceutical enterprises in the group is higher than the threshold, some pharmaceutical enterprises choose the R&D innovation strategy, and the two strategies coexist in the group. It can be seen from Figure 3B that when the R&D success rate is low, pharmaceutical enterprises always tend to choose the non-R&D innovation strategy.

### 6.4. Comparing Expected Revenue and Super Expected Revenue

Let ξ=1.5, $\eta_{1}=0.7$, $\eta_{2}=0.3$, and $g_{i}=f_{i}-1$. To analyze the differences in the strategic choices under the dominance of expected revenue and those under the dominance of super expected revenue, Figure 4 shows the changes in the h of strategic choice under two different conditions.

Figure 4A shows the impact of the number of pharmaceutical enterprises on revenue, and Figure 4B shows the impact of tax incentives on revenue. Under the dominance of expected revenue and the dominance of super expected revenue, the critical conditions for strategic choice are the same, and the strategic choice of pharmaceutical enterprises is also the same. Expected revenue and super expected revenue do not affect the strategic choice. However, under the dominance of super expected revenue, pharmaceutical enterprises are more quickly affected by revenue.

## 7. Conclusions

Based on the Moran process, this paper studies how pharmaceutical enterprises choose the R&D innovation strategy and the non-R&D innovation strategy based on changes in tax incentives, R&D success rate, and the number of pharmaceutical enterprises under the dominance of external factors, expected revenue, and super expected revenue. The strategic choice change process was simulated by MATLAB 2021b. The main conclusions and recommendations were as follows.

Firstly, the number of pharmaceutical enterprises in the market has an impact on strategic choices. The higher the number of pharmaceutical enterprises, the lower the probability of pharmaceutical enterprises choosing the R&D innovation strategy. Leading pharmaceutical enterprises in the industry should adhere to their social responsibilities, actively participate in drug R&D innovation, and drive the enthusiasm of small- and medium-sized enterprises to innovate.

Secondly, government departments provide appropriate tax incentives for R&D and innovative enterprises, which is conducive to promoting pharmaceutical enterprises to choose the R&D innovation strategy. Therefore, the government departments should give different tax incentives to enterprises of different scales and different production capacities. Through formulation and improvement, the preferential tax policies can be better-matched with the current situation of pharmaceutical enterprises’ R&D, and the policy costs of R&D by pharmaceutical enterprises can be further reduced.

Thirdly, the improvement of the R&D success rate is conducive to promoting pharmaceutical enterprises to choose the R&D innovation strategy. Therefore, pharmaceutical enterprises should increase investment in R&D, focus on the cultivation and introduction of internal high R&D talents, comprehensively optimize R&D activities, and improve the success rates of drug R&D.

Finally, the participation of pharmaceutical enterprises in industry cooperation helps improve the R&D level. Therefore, pharmaceutical enterprises should actively participate in the R&D of industry collaborative innovation and give full play to the advantages of multi-subject participation. All participants should create a good environment for industry cooperation, build a professional service platform, and create a good innovation environment to guide the R&D and innovation of pharmaceutical enterprises.

The research can be further expanded. Firstly, the multiple strategic choices of pharmaceutical companies can be considered, and secondly, random processes can be introduced to subdivide the external factors that affect strategic choices.

## Figures and Tables

**Figure 1 ijerph-19-15197-f001:**
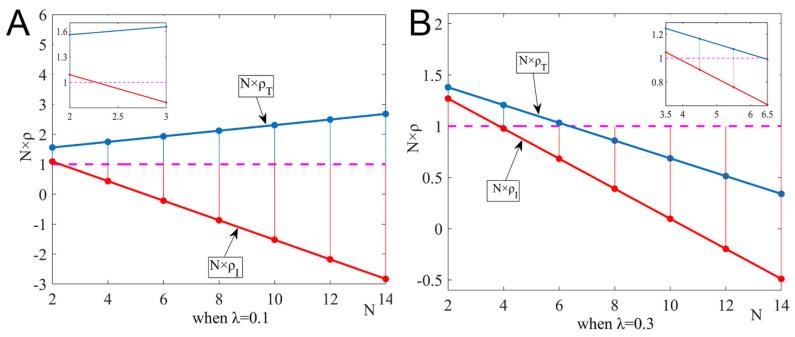
Impact of the number of pharmaceutical enterprises on fixed-point probability when the government subsidy rate is 0.1 and 0.3. (**A**) When λ=0.1, and (**B**) when λ=0.3.

**Figure 2 ijerph-19-15197-f002:**
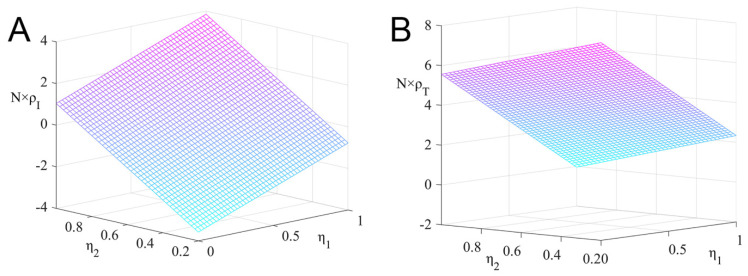
The impact of R&D success rate on the fixed-point probability of the R&D innovation strategy and the non-R&D innovation strategy under the dominance of external factors. (**A**) Fixed point probability $N\times \rho_{I}$, and (**B**) fixed point probability $N\times \rho_{T}$.

**Figure 3 ijerph-19-15197-f003:**
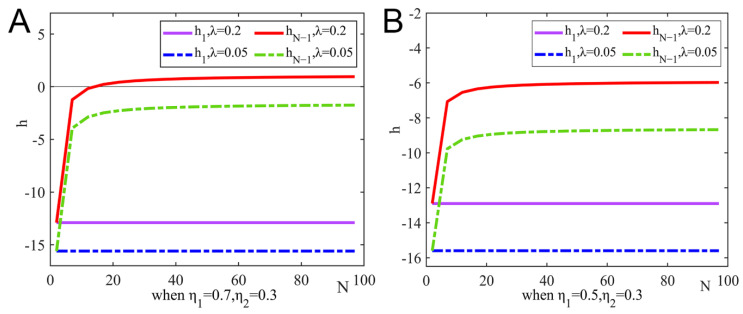
The impact of the number of pharmaceutical enterprises and tax incentives on strategic choice under the dominance of expected revenue. (**A**) When $\eta_{1}=0.7,\eta_{2}=0.3$, and (**B**) when $\eta_{1}=0.5,\eta_{2}=0.3$.

**Figure 4 ijerph-19-15197-f004:**
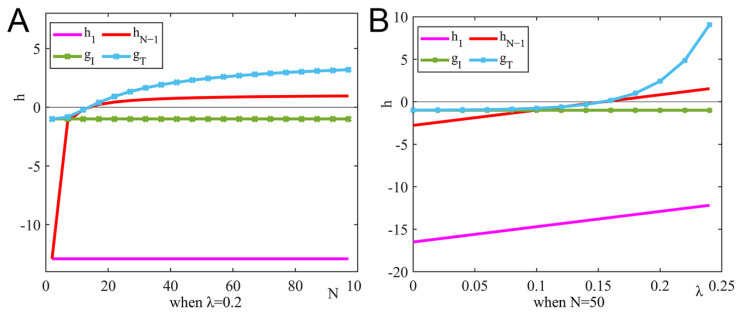
Strategy evolution process under different conditions. The impact of the number of pharmaceutical enterprises and government tax incentives on strategic choice under the dominance of expected revenue and super expected revenue. (**A**) When λ=0.2, and (**B**) when N=50.

**Table 1 ijerph-19-15197-t001:** Strategy revenues matrix of pharmaceutical enterprises.

		Pharmaceutical Enterprises (Q)	
		R&D Innovation (I)	Non-R&D Innovation (T)
Pharmaceutical enterprises (P)	R&D innovation (I)	$\eta_{1}R_{I}-C_{ph}$ $\eta_{1}R_{I}-C_{ph}$	$\eta_{2}R_{I}-C_{ph}$ $R_{T}-\lambda R_{T}-C_{pl}$
	Non-R&D innovation (T)	$R_{T}-\lambda R_{T}-C_{pl}$ $\eta_{2}R_{I}-C_{ph}$	$R_{T}-\lambda R_{T}-C_{pl}$ $R_{T}-\lambda R_{T}-C_{pl}$

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
