# Peer review of "Pharmaceutical Enterprises’ R&D Innovation Cooperation Moran Strategy When Considering Tax Incentives"

_ijerph, 2022, doi:10.3390/ijerph192215197_

Round 1

Reviewer 1 Report

Title: Pharmaceutical Enterprises R&D Innovation Cooperation Moran Strategy When Considering Tax Incentives

This manuscript looks to describe Pharmaceutical Firms strategic choice betwen R&D Innovations and Non-R&D innovations using the Evolutionary game dynamics of Moran process, taking industry-university-research cooperation under tax subsidies into consideration. 

In really, the R&D investment decisions of the pharmaceutical companies are directly determined by expected revenue from a new drug as well as the expected cost of the R&D development, and the competitions between pharmaceutical companies in competitive/monopoly/oligoplay markets, plus public policies that influence the supply of and demand for drugs. These above-mentioned market forces based on the economic theories seem not include the factor mentioned in this manuscript: the tax subsidies on university-industry collaboration. 

Big challenges come from the Model Assumptions on which the Moran process was constructed.

 Line 177, the two types of participants {P,Q} need to be clearly defined.

 Line 183, it is not clear why the assumption may hold that RI > RT >0

Lines 186-189, Assumption H3, it is not clear why the it is assumed that industry-university-research cooperative innovation organization may improve the R&D successful rates? Is there any empirical evidence available from the global or regional markets?

Lines 193-197, could you please cite previous literatures regarding tax exemption on pharmaceutical companies’ innovative R&D activities?

Line 200, Table I is confusing. The benefit matrix reflects two enterprises instead of N enterprises. 

Lines 222- 233, the variable ξ of the selection strength need to be explained in details. Why when ξ >1, dominance of exceed expected returns applies?

Line 231, in the model (4), which ξ is still assumed between 0 and 1?

Given that the assumptions lack empirical evidences and deviate from traditional economic theory of pharmaceutical drugs markets, it is challengeable for the reviewer to comment on the model simulation results based on the Moran process.

In the section 6, Could the authors may think of a Mablab simulation plots of the Impacts of the selection strengths, tax and subsidies on the R&D successful rates(a 3 Dimensional plot), Since ξ of the selection strength is continuous?

Finally, the conclusion and discussion sections may be merged into one section “Conclusion and Discussion”

Author Response

Responses to Reviewer 1

Dear Reviewer,

Thank you very much for giving us an opportunity to submit a revised version of the manuscript entitled “Pharmaceutical Companies R&D Innovation Cooperation Mo-ran Strategy When Considering Tax Incentives”. The manuscript was revised submission (Manuscript ID: ijerph-1844920) with new line and page numbers in the text. The revised paper contains 17 pages, 4 figures and 1 table.

We are also thankful to you for your critical reading and valuable comments on the manuscript. Those comments point out the direction for our further study. The relevant changes had been made in the original manuscript according to the comments, and the major revised portions were marked in red. We also responded to your comments one by one, along with a clear indication of the location of the revision.

Regarding language, we apologize for the lack of language in the manuscript. We worked on the manuscript for a long time, and repeated revisions of sentences and chapters could lead to poor readability of the article. We make improvements in the manner suggested by the editors, and we hope that the overall article structure and language have improved considerably.

In addition, after careful consideration, we have further improved the paper without changing the structure and general content of the paper.

Comment #1:

In really, the R&D investment decisions of the pharmaceutical companies are directly determined by expected revenue from a new drug as well as the expected cost of the R&D development, and the competitions between pharmaceutical companies in competitive/monopoly/oligoplay markets, plus public policies that influence the supply of and demand for drugs. These above-mentioned market forces based on the economic theories seem not include the factor mentioned in this manuscript: the tax subsidies on university-industry collaboration. 

Response: Thank you very much for your careful review work. This paper analyzes the strategy choice of pharmaceutical companies by constructing a Moran process random evolution game model. As pointed out in the opinion, we study the R&D innovation decisions of pharmaceutical companies based on preferential tax policies and the industry-university-research cooperation mechanism.

We believe that tax incentives will help boost the R&D of pharmaceutical companies. The reason is that drug R&D has the characteristics of high risk and large investment, and the economic pressure of R&D investment of pharmaceutical companies is gradually increasing. Reducing taxes through reasonable planning can reduce the expected cost of R&D for pharmaceutical companies, thereby effectively encouraging pharmaceutical companies to invest in R&D. Jia J et al. also conducted research on tax subsidies and R&D investment, and found that tax incentives are effective in stimulating R&D activities.

Regarding industry-university-research cooperation, the participation of pharmaceutical companies in industry-university-research cooperation can effectively improve the efficiency of enterprise R&D investment. Pharmaceutical companies cooperate with universities and research institutes to conduct special research on key technologies, break through research problems, and promote drug research and development. In terms of industry-university-research cooperation to promote R&D innovation, Li M et al. (2021)[1] have proved through research that cooperation between companies and other innovative entities will help improve their R&D capabilities; Zhao J et al. (2017)[2] found through data analysis that industry-university-research collaboration can help promote innovation capabilities.

Therefore, we use tax subsidies and industry-university-research cooperation as influencing factors to analyze their impact on pharmaceutical companies' R&D decisions.

Comment #2:

Line 177, the two types of participants {P, Q} need to be clearly defined.

Response: Thank you very much for the suggestion. In the population of pharmaceutical companies, there is a 2*2 symmetrical game between R&D innovation strategies and non-R&D innovation strategies. Among them, the number of pharmaceutical companies that choose R&D innovation strategies is, and the number of pharmaceutical companies that choose non-R&D innovation strategies is. In the Moran process, pharmaceutical companies make a strategic choice at each time step. In each strategy selection, two pharmaceutical companies are randomly selected from N pharmaceutical companies to play against each other. In order to distinguish the two pharmaceutical companies in the game, we define them as {pharmaceutical company P, pharmaceutical company Q} respectively. We make a supplementary explanation on line 198-200.

Comment #3:

Line 183, it is not clear why the assumption may hold that RI > RT >0.

Response: Thank you very much for your careful review work. We assume that the pharmaceutical company's revenue from selling R&D and innovative drugs is, and the revenue from selling drugs that have not R&D and innovative production is, . Because pharmaceutical companies need to continuously invest in R&D to independently develop drugs. R&D and innovation improve the quality of medicines and drive the income of pharmaceutical companies. Quiroz K (2016)[3] emphasizes R&D as a strategy to increase revenue by analyzing merger cases of pharmaceutical companies. Guo Y et al. (2021)[4] hypothesized that the company's R&D innovation gains additional income, that is, the R&D innovation income is higher than the non-R&D innovation income. Therefore, in order to distinguish the benefits of R&D innovation activities and non-R&D innovation activities, we assume.

Comment #4:

Lines 186-189, Assumption H3, it is not clear why the it is assumed that industry-university-research cooperative innovation organization may improve the R&D successful rates? Is there any empirical evidence available from the global or regional markets?

Response: Thank you very much for the suggestion. The industry-university-research cooperation model can give full play to the inherent advantages of multiple parties and enhance the overall R&D strength. Pharmaceutical companies can make full use of the scientific research advantages of scientific research institutes to conduct special research on key technologies for drug research and development. By applying these research results to the process of drug research and development, the difficulties in the process of drug research and development can be solved, and the success rate of drug research and development can be improved.

At present, in terms of industry-university-research cooperation to promote R&D innovation, Li M et al. (2021)[1] have proved through research that cooperation between companies and other innovative entities will help provide continuous technical support for their technological innovation, thereby enhancing their R&D capabilities. Zhao J et al. (2017)[2] collected the indicators and data affecting China's industry-university-research cooperation innovation system from 2005 to 2014, and used MATLAB software to simulate the evolution of related variables, and found that industry-university-research collaboration can help promote innovation capabilities.

Therefore, we hypothesize that the participation of pharmaceutical companies in industry-university-research cooperation can help improve the success rate of R&D. We cited the supplementary description of the relevant literature in 2.2 R&D innovation cooperation (Modified line 124-126).

Comment #5:

Lines 193-197, could you please cite previous literatures regarding tax exemption on pharmaceutical companies’ innovative R&D activities?

Response: Thank you very much for your careful review work. Drug research and development requires a lot of capital investment. In order to promote drug research and development, government departments have issued a series of tax reduction and exemption policies. Therefore, we consider the impact of government tax incentives on pharmaceutical companies' R&D strategies. We assume that for pharmaceutical companies that choose R&D strategies, the government department reduces some of their taxes. We state it as "tax-exempt", which is our logical mess. We have re-described the hypothetical content of this section (modified line 191-196).

Comment #6:

Line 200, Table I is confusing. The benefit matrix reflects two companies instead of N companies. 

Response: Thank you very much for your careful review work. We name Table 1 as "N (finite field) pharmaceutical companies' production strategy return matrix". N pharmaceutical companies represent a total of N pharmaceutical companies in the market. The income matrix represents that at each update step, two pharmaceutical companies are randomly selected from N populations to play against each other. Therefore, the income matrix reflects the game result of choosing two pharmaceutical companies from N pharmaceutical companies. The revenue matrix is just two firms. In order to express this game matrix more clearly, we modify Table 1 as "Production Strategy Benefit Matrix between Pharmaceutical Companies" (modified line 201).

Comment #7:

Lines 222- 233, the variable ξ of the selection strength need to be explained in details. Why when ξ >1, dominance of exceed expected returns applies?

Response: Thank you very much for your careful review work. Referring to the research of Taylor C et al. (2004)[5], when the selection strength parameter, the participant effect function based on the Moran process of the stochastic evolutionary game is in the form of a linear function:

It represents the strategic choice strength of pharmaceutical companies under the two scenarios dominated by random factors and those dominated by expected returns. According to Traulsen A et al. (2008)[6], for strong selection processes, there is an upper limit on the selection strength of linear functions. So we further assume that relative fitness is a nonlinear function of exponential form:

When, the negative payoff causes the relative fitnessto be close to 0, while the positive payoff causes the relative fitness to be very large. Since the exponential function is positive for any argument, negative and positive entries in the payoff matrix can be analyzed without restrictions. The relative fitnesswill vary greatly when payoff changes. Therefore, when, we believe that the beat-to-earth returns dominate.

Comment #8:

Line 231, in the model (4), which ξ is still assumed between 0 and 1?

Response: Thank you very much for your careful review work. We are very sorry for our incorrect writing and it is rectified at line. The correct formula (4) should be "" (modified line231). The parameterrepresents the selection strength and can take any positive value, but we only calculate the excess expected return part through Equation 4, so only the case ofis analyzed.

Comment #9:

In the section 6, Could the authors may think of a Mablab simulation plots of the Impacts of the selection strengths, tax and subsidies on the R&D successful rates(a 3 Dimensional plot), Since ξ of the selection strength is continuous?

Response: We thank you for reading our manuscript and for providing us with advice. We tried to draw this picture according to your suggestion, but we couldn't finish it.

First, the selection strengthis continuous, but we have divided it into three parts for our analysis. When, random factors dominate, we judge the strategic choice of pharmaceutical companies by analyzing the fixed point probability; when, the expected return dominates, and we judge the strategy choice of pharmaceutical companies by analyzing the differencebetween the utility functions of the two strategies at different times; when, the super-expected return dominates, and we judge the strategic choice of pharmaceutical companies by analyzing the utility function ratioof the two strategies at different times. In each scenario, the selection intensity has a different effect on strategy, and we cannot precisely define the scope of in the three scenarios.

Second, to analyze the impact of selection intensityand tax subsidies rateon the success rate of R&D , it is necessary to use a fixed formula to link them. We cannot set the value of, so we cannot judge the impact of selective intensity and tax subsidies on the success rate of R&D. Therefore, the drawing of this graph cannot be completed.

Comment #10:

Finally, the conclusion and discussion sections may be merged into one section “Conclusion and Discussion”

Response: Thank you very much for your suggestion. We have read the “Discussion” and “Conclusions” sections carefully and merged them into the “Conclusions and Future Work” section (revised page15-16). We are more than happy to revise the article further based on the valuable comments of the reviewers.

References:

  • Li M, Zhang M, Agyeman F O. Research on the influence of industry-university-research cooperation innovation network characteristics on subject innovation performance[J]. Mathematical Problems in Engineering, 2021, 2021. doi:1155/2021/4771113.
  • Zhao J, Wu G. Evolution of the Chinese Industry-University-Research collaborative innovation system[J]. Complexity, 2017, 2017. doi:1155/2017/4215805.
  • Quiroz K. Pharmaceutical megamergers’ dependence on existing products: The case for R&D in the Pfizer-Allergan merger[J]. Strategic Direction, 2016, 32(6): 30-32. doi:1108/SD-03-2016-0041.
  • Guo Y, Zou H, Liu Z. Behavioral analysis of subjects for green technology innovation: a tripartite evolutionary game model[J]. Mathematical Problems in Engineering, 2021, 2021. doi:1155/2021/5181557.
  • Taylor C, Fudenberg D, Sasaki A, et al. Evolutionary game dynamics in finite populations[J]. Bulletin of mathematical biology, 2004, 66(6): 1621-1644. doi:10.1016/j.bulm.2004.03.004.
  • Traulsen A, Shoresh N, Nowak M A. Analytical results for individual and group selection of any intensity[J]. Bulletin of mathematical biology, 2008, 70(5): 1410-1424. doi:10.1007/s11538-008-9305-6.

We have tried our best to improve the manuscript and made some changes according to your comments. We earnestly appreciate your professional work and hope that the corrections will make our manuscript suitable for publication. We are looking forward to receiving comments in the future. If you have any questions, please do not hesitate to contact me at the address below.

Once again, thank you very much for your valuable comments and suggestions.

Best wishes.

Sincerely,

Lilong Zhu, Ph.D. and Professor. Postdoctoral research in Shandong University. Research scholar in College of business, University of Illinois at Urbana-Champaign, USA. Ph.D. degree was granted in management science and engineering from Tongji University. His research interests are supply chain management and quality management.

E-mail: zhulilong2008@126.com, zhulilong@sdnu.edu.cn

Tel: +86-13853193366

School of Business, Shandong Normal University, Jinan 250014, Shandong, China.

Reviewer 2 Report

The authors focused on an issue that is very important to public health and health policy in all countries. The covid-19 pandemic and international / global conflicts have shown that the globalization of drug production and supply chains pose a threat to the health of individual societies. Therefore - as suggested by the authors - each country should strive to develop its own pharmaceutical industry. The authors focused on selected factors stimulating this development, in particular R&D and tax incentives.

The article is theoretical and the analysis uses the Moran stochastic evolution game model. The applied analytical tool is well selected and allows to obtain results of significance for empiricism.

The article is written clearly, the individual stages of the analysis are supported by mathematical proofs, and for interested readers without proper mathematical preparation, the authors provided explanations and comments. It is very important as such an analysis stimulates verification with the use of empirical research.

The introduction contains selected elements of the research background, there is no introduction directly related to the problem being the subject of the analysis.

Similarly, the conclusions presented a general summary without referring to the results obtained. In the individual points of the article, all the necessary results, explanations and discussions are included. However, it is worth including a summary "in a nutshell" in the conclusions, which would help, for example, when designing empirical research.

Author Response

Responses to Reviewer 2

Dear Reviewer,

Thank you very much for giving us an opportunity to submit a revised version of the manuscript entitled “Pharmaceutical Enterprises R&D Innovation Cooperation Mo-ran Strategy When Considering Tax Incentives”. The manuscript was revised submission (Manuscript ID: ijerph-1844920) with new line and page numbers in the text. The revised paper contains 17 pages, 4 figures and 1 table.

We are also thankful to you for your critical reading and valuable comments on the manuscript. Those comments point out the direction for our further study. The relevant changes had been made in the original manuscript according to the comments, and the major revised portions were marked in red. We also responded to your comments one by one, along with a clear indication of the location of the revision. In addition, after careful consideration, we have further improved the paper without changing the structure and general content of the paper.

Comment #1:

The introduction contains selected elements of the research background, there is no introduction directly related to the problem being the subject of the analysis.

Response: Thank you very much for your suggestion. According to your suggestion, we have supplemented the introduction part, mainly adding the part related to the problem under analysis (modified line 64-69). We have supplemented the role of tax incentives and industry-university-research cooperation in the R&D activities of pharmaceutical enterprises. The supplementary content is as follows:

In this context, the government departments have issued a series of preferential tax policies to increase the willingness of pharmaceutical enterprises to R&D. Pharmaceutical enterprises are encouraged to actively cooperate with universities and re-search institutes to improve drug R&D technology. Therefore, considering tax incentives and industry-university-research cooperation is of great practical significance for promoting drug R&D.

Comment #2:

Similarly, the conclusions presented a general summary without referring to the results obtained. In the individual points of the article, all the necessary results, explanations and discussions are included. However, it is worth including a summary "in a nutshell" in the conclusions, which would help, for example, when designing empirical research.

Response: Thank you very much for your suggestion. In the discussion in Section VII, we analyze and recommend the results of the model analysis. Therefore, in the conclusion of Section VIII, we just elaborate the main structure of the article and future research directions. We thought about your suggestion and decided to combine Parts 7 and 8 into one section, "Conclusions and future works" (revised page 15-16). In this way, the last section both refers to the results obtained and provides a general summary of the article.

We have tried our best to improve the manuscript and made some changes according to your comments. We earnestly appreciate your professional work and hope that the corrections will make our manuscript suitable for publication. We are looking forward to receiving comments in the future. If you have any questions, please do not hesitate to contact me at the address below.

Once again, thank you very much for your valuable comments and suggestions.

Best wishes.

Sincerely,

Lilong Zhu, Ph.D. and Professor. Postdoctoral research in Shandong University. Research scholar in College of business, University of Illinois at Urbana-Champaign, USA. Ph.D. degree was granted in management science and engineering from Tongji University. His research interests are supply chain management and quality management.

E-mail: zhulilong2008@126.com, zhulilong@sdnu.edu.cn

Tel: +86-13853193366

School of Business, Shandong Normal University, Jinan 250014, Shandong, China.

Reviewer 3 Report

Pharmaceutical Enterprises R&D Innovation Cooperation Moran Strategy When Considering Tax Incentives

Although the idea is worth exploring, the manuscript shows severe flaws of which the most important remarks are discussed below. For that reason, the paper cannot be considered for publication.

1.      Positioning and contribution

Little attention is paid to explain in what respect your study covers some important gaps in the literature. For example, stating that applying a Moran process random evolutionary game model aims to solve three aspects determining pharmaceutical enterprises’ R&D strategic choices (elaborated in lines 73-79)  is not convincing, nor does the enumeration explains how it contributes to current findings/approaches in the literature. What is investigated in this respect so far? In addition, several paragraphs in the introduction have no references/sources, which is of course problematic. Some examples and figures are given to illustrate the increasing emphasis on R&D innovation without any reference (see lines 41-53). Also, the statement that the government sector is insufficient to guide with incentives is not founded in the literature (line 59). Moreover, a general remark on the manuscript is the need to explain to what extent the research is valid to a specific institutional context or certain types of countries. In terms of the practical implications this is important to explain.

2.      Literature review

The literature review is too little related to your research question and needs more focus and elaboration. Section 2.1, for example, mentions the drugs quality of the pharmaceutical enterprises. However, in the paragraph is not clear what is meant with ‘quality’. Intuitively, I would relate it to customer satisfaction, quality control, failure rate,…However, the studies mentioned in the section mainly deal with R&D costs and risks. Also explain why it is relevant for the innovation strategy choices investigated in your study.  

3.      Model assumptions

Several concepts for which assumptions are made in the model are not clear to me. In particular, I wonder how these concepts are defined and what might they imply in reality. For example, each pharmaceutical enterprise has two strategies to choose from: an R&D innovation strategy and a non-R&D innovation strategy. How do you characterize a non-R&D innovation strategy? At least this should be discussed in the literature section. Moreover, what is meant with the R&D success rate? There could be several ways to look at it; number of new drugs, number of patents,… So explain why you consider the income of drugs in your model. Another question relates to the choice to measure tax savings as lambda times the sales revenue of non-R&D innovative drugs. Besides income-based R&D tax incentives also expenditure-based tax reliefs exist in practice (OECD, 2016). Also, make clear what kind of external random factors could have an impact on the strategy selection of pharmaceutical enterprises (line 219).

4.      Simulation Analysis

Please explain why the production cost is 13 and the profit 22 if pharmaceutical enterprises can choose the original production mode (lines 407-411). I am afraid I cannot follow there.

5.      Conclusions and future works

In general, the policy implications mentioned in the conclusions are vague and sloppy. It is mentioned that that the government should supervise the capitalization of R&D (balance sheet) as the could be used to manipulate profits. However, this could also be the case for expensing R&D on the income statement. I do not understand the message here.

6.      Language

The level of proficiency in the English language is rather poor. Asking a (native) colleague or using a professional language editing service could be helpful to ensure that your meaning is clear and identify problems that require your review. Problems occur regarding grammar, sentence structure and spelling. Some examples:

… are of great significance to promote innovation and development (line 10)

…the number of pharmaceutical enterprises w be discussed (line 412)

…The success rate of…., far exceeding the industry average (line 49-50): grammatically not correct

Reference:

·        Appelt, S. et al. (2016-09-10), “R&D Tax Incentives: Evidence on design, incidence and impacts”, OECD Science, Technology and Industry Policy Papers, No. 32, OECD Publishing, Paris. http://dx.doi.org/10.1787/5jlr8fldqk7j-en

Author Response

Dear Editor and Reviewer 3,

Thank you very much for giving us an opportunity to submit a revised version of the manuscript entitled “Pharmaceutical Companies R&D Innovation Cooperation Mo-ran Strategy When Considering Tax Incentives”. The manuscript was revised submission (Manuscript ID: ijerph-1844920) with new line and page numbers in the text. The revised paper contains 15 pages, 4 figures and 1 table.

We are also thankful to the reviewer for their critical reading and valuable comments on the manuscript. Those comments point out the direction for our further study. The relevant changes had been made in the original manuscript according to the comments of reviewers, and the major revised portions were marked in red. We also responded to the reviewer's comments one by one, along with a clear indication of the location of the revision, the specific content is in “Responses to Reviewer 3”. In addition, after careful consideration, we have further improved the paper without changing the structure and general content of the paper.

In addition, after careful consideration, we have further improved the paper without changing the structure and general content of the paper.

Comment #1:  Positioning and contribution

Little attention is paid to explain in what respect your study covers some important gaps in the literature. For example, stating that applying a Moran process random evolutionary game model aims to solve three aspects determining pharmaceutical enterprises’ R&D strategic choices (elaborated in lines 73-79) is not convincing, nor does the enumeration explains how it contributes to current findings/approaches in the literature. What is investigated in this respect so far? In addition, several paragraphs in the introduction have no references/sources, which is of course problematic. Some examples and figures are given to illustrate the increasing emphasis on R&D innovation without any reference (see lines 41-53). Also, the statement that the government sector is insufficient to guide with incentives is not founded in the literature (line 59). Moreover, a general remark on the manuscript is the need to explain to what extent the research is valid to a specific institutional context or certain types of countries. In terms of the practical implications this is important to explain.

Response: Thank you very much for your careful review work. We divide your comment and explain them separately.

(1) Thank you very much for pointing this out. At present, some scholars have studied the impact of tax incentives and industry-university-research cooperation on the R&D innovation through empirical models. Dai et al. [1] believe that tax incentives have recently become a popular option in many countries as part of a policy to stimulate R&D activities. Furthermore, innovation, as a consequence, will improve. Bai X J et al. [2] base on the dynamic network slacks-based measurement model analyzes the importance of industry-university-research cooperation to innovation. However, no scholars have analyzed the impact of tax incentives and industry-university-research cooperation on the strategic choice of pharmaceutical enterprises by constructing a stochastic evolutionary game model. The Moran process subdivides the market environment into external-factors-dominant, expected-revenue-dominant and super-expected-revenue-dominant, and discusses the evolutionary process of the strategic choice of pharmaceutical enterprises respectively. There are few studies on this aspect at present. Therefore, this paper builds an evolutionary game model based on the Moran process to solve the impact of tax incentives and industry-university-research cooperation on strategic choice under the dominance of external factors, the dominance of expected revenue, and the dominance of super expected revenue.

(2) In March 2022, "Fierce Biotech" released the list of the top ten pharmaceutical companies with R&D investment in the world in 2021. Our data comes from this leaderboard. In the new manuscript, we explain the source of the paper's data. Please see page 1 of the revised manuscript, lines 33-35. From the data, we can see that pharmaceutical enterprises are increasing their investment in R&D activities and attaching importance to R&D production.

(3) We thank the reviewer for pointing this out. At present, governments around the world have introduced many incentives to promote the R&D of pharmaceutical enterprises. After reviewing relevant literature, we decided to delete "the government sector is insufficient to guide with incentives". In the new manuscript, we have revised this part. Please see page 1 of the revised manuscript, lines 43-44, page 2, lines 45-48.

(4) We think this is an excellent suggestion. We analyze the impact of government tax incentives and industry-university-research cooperation on the choice of pharmaceutical companies' R&D strategies. Most governments, especially developing countries, have implemented tax incentives to promote foreign investment and stimulate economic development in their country. These initiatives offer more favorable tax treatment to specific economic activities [3]. China, Singapore, India, Ireland, and other sites also offer attractive tax incentives for pharmaceutical manufacturing [4]. Our research provides recommendations for these countries with tax incentives.

Comment #2:  Literature review

The literature review is too little related to your research question and needs more focus and elaboration. Section 2.1, for example, mentions the drugs quality of the pharmaceutical enterprises. However, in the paragraph is not clear what is meant with ‘quality’. Intuitively, I would relate it to customer satisfaction, quality control, failure rate,…However, the studies mentioned in the section mainly deal with R&D costs and risks. Also explain why it is relevant for the innovation strategy choices investigated in your study.  

Response: Thank you very much for your careful review work. We have supplemented and clarified the literature review as you suggested and hope to now meet the requirements. Please see page 2 of the revised manuscript, lines 66-86. In Section 2.1, we mainly address the costs and risks of R&D innovation. This is because the higher costs and risks are important factors affecting the innovation strategic choice. The pharmaceutical enterprises’ R&D helps to improve the drug quality, but the drug quality is not the focus of this paper. We mistakenly named Section 2.1 "drug quality", which has been changed to "R&D innovation and government incentives" in the new manuscript. We hope that the new literature review will be more relevant to the paper.

Comment #3: Model assumptions

Several concepts for which assumptions are made in the model are not clear to me. In particular, I wonder how these concepts are defined and what might they imply in reality. For example, each pharmaceutical enterprise has two strategies to choose from: an R&D innovation strategy and a non-R&D innovation strategy. How do you characterize a non-R&D innovation strategy? At least this should be discussed in the literature section. Moreover, what is meant with the R&D success rate? There could be several ways to look at it; number of new drugs, number of patents, So explain why you consider the income of drugs in your model. Another question relates to the choice to measure tax savings as lambda times the sales revenue of non-R&D innovative drugs. Besides income-based R&D tax incentives also expenditure-based tax reliefs exist in practice (OECD, 2016). Also, make clear what kind of external random factors could have an impact on the strategy selection of pharmaceutical enterprises (line 219).

Response: Thank you very much for your careful review work. We divide your comment and explain them separately.

(1) We divide the strategies of pharmaceutical enterprises into "R&D innovation strategy" and "non-R&D innovation strategy". The R&D innovation strategy means that pharmaceutical enterprises increase investment in the R&D of existing drugs to improve the efficacy. The non-R&D innovation strategy refers to the continuous production of drugs according to the existing production methods, without additional R&D investment, technological improvement, etc. We have supplemented in the new manuscript. Please see page 3 of the revised manuscript, lines 144-146, and page 4, lines147-148.

(2) In this paper, we consider R&D success rate means the number of drugs. The increase in the R&D investment help improve the profitability of enterprises (Tung et al. 2022) [5]. The success rate of R&D of pharmaceutical companies will affect the opportunity income of the company. In order to measure the revenue of R&D to pharmaceutical companies, we combine that with revenue.

(3) The current tax incentives include expenditure-based tax support, income-based tax support, etc (Appelt et al. 2016) [6]. We have studied the papers on tax incentives in recent years and found that most of them are related to expenditure to analyze the impact on corporate activities, and fewer papers are related to income to analyze problems. Therefore, this paper links the tax incentive rate with income, and analyzes the impact of income-based tax model on the R&D strategy of pharmaceutical enterprises.

(4) Thank you very much for your suggestion. The external factors that affect the innovation strategy of pharmaceutical enterprises mainly refer to policy factors, market demand, and corporate social responsibility. This is our next research direction, subdividing external factors and studying their impact on enterprises strategic choices.

Comment #4: Simulation Analysis

Please explain why the production cost is 13 and the profit 22 if pharmaceutical enterprises can choose the original production mode (lines 407-411). I am afraid I cannot follow there.

Response: Thank you very much for your careful review work. We apologize for not making the strategy clear. The original production mode means that pharmaceutical enterprises choose non-R&D innovation strategies and maintain the original production mode. In the new manuscript, we have changed it to "When pharmaceutical enterprises choose non-R&D innovation strategy".

Regarding the production cost being 13 and the profit being 22, this is a value determined by us after many calculations. We simulate the strategic choice process. We refer to the cost and profit in reality, and calculate the cost and profit with different values. After several simulations, we found that 13 and 22 are more in line with our model. Therefore, we take the cost as 13 and the profit as 22.

Comment #5: Conclusions and future works

In general, the policy implications mentioned in the conclusions are vague and sloppy. It is mentioned that that the government should supervise the capitalization of R&D (balance sheet) as the could be used to manipulate profits. However, this could also be the case for expensing R&D on the income statement. I do not understand the message here.

Response: Thank you very much for your careful review work. We apologize for the vague policy by the previous version of the manuscript and hope the new policy is more realistic. Regarding the question " It is mentioned that that the government should supervise the capitalization of R&D (balance sheet) as the could be used to manipulate profits. However, this could also be the case for expensing R&D on the income statement.", we have removed. We reformulate our recommendations based on the model analysis results. Please see page 13 of the revised manuscript, lines 412-445.

Comment #6: Language

The level of proficiency in the English language is rather poor. Asking a (native) colleague or using a professional language editing service could be helpful to ensure that your meaning is clear and identify problems that require your review. Problems occur regarding grammar, sentence structure and spelling. Some examples:

… are of great significance to promote innovation and development (line 10)

…the number of pharmaceutical enterprises w be discussed (line 412)

…The success rate of…., far exceeding the industry average (line 49-50): grammatically not correct

Response: Thank you very much for your careful review work. We apologize for the poor language and careless mistakes of our manuscript. We worked on the manuscript for a long time and the repeated addition and removal of sentences and sections obviously led to poor readability. For the three examples, we make careful modifications. We have now worked on both language and readability and have also involved native English speakers for language corrections. We really hope that the flow and language level have been substantially improved.

We would like to thank the referee again for taking the time to review our manuscript. We hope that our responses and edits are satisfactory.

  • Dai, X., Verreynne, M., Wang, J. and He, Y. (2020), “The behavioral additionality effects of a tax incentive program on firms' composition of R&D investment”, R&D Management, Wiley, Vol. 504, pp. 510-521. doi:10.1111/radm.12401
  • Bai X J, Li Z Y, Zeng J. Performance evaluation of China's innovation during the industry-university-research collaboration process—an analysis basis on the dynamic network slacks-based measurement model. Technology in Society, 2020, 62: 101310. doi:10.1016/j.techsoc.2020.101310
  • Klemm A, Van Parys S. Empirical evidence on the effects of tax incentives. Int Tax Public Financ. 2012;19(3):393–423. :10.1007/s10797-011-9194-8.
  • Fontalvo-Lascano M A, Alvarado-Hernández B B, Conde C, et al. Development and Application of a Business Case Model for a Stream Sampler in the Pharmaceutical Industry. Journal of Pharmaceutical Innovation, 2022,2022: 1-13. doi:10.1007/s12247-022-09634-0
  • Tung L T, Binh Q M Q. The impact of R&D expenditure on firm performance in emerging markets: evidence from the Vietnamese listed companies. Asian Journal of Technology Innovation, 2022, 30(2): 447-465. doi:10.1080/19761597.2021.1897470
  • Appelt, S. et al. (2016-09-10), “R&D Tax Incentives: Evidence on design, incidence and impacts”, OECD Science, Technology and Industry Policy Papers, 32, OECD Publishing, Paris.  doi:10.1787/5jlr8fldqk7j-en

Best wishes.

Sincerely,

Lilong Zhu, Ph.D. and Professor.

Postdoctoral research in Shandong University. Research scholar in College of business, University of Illinois at Urbana-Champaign, USA. Ph.D. degree was granted in management science and engineering from Tongji University and Postdoctoral research in Shandong University. His research interests are supply chain management and quality management.

E-mail: zhulilong2008@126.com, zhulilong@sdnu.edu.cn

Tel: +86-531-13853193366

College of Business, Shandong Normal University, Ji’nan 250014, Shandong, China.

Round 2

Reviewer 3 Report

Dear authors,

Thank you for your clear responses to my suggestions and remarks.